# Two-Steps Diffusion Policy for Robotic Manipulation via Genetic Denoising

**Mateo Clemente\***
Huawei Technologies Canada
mateo.clemente@h-partners.com

**Léo Maxime Brunswic\***
Huawei Technologies Canada
leo.maxime.brunswic@h-partners.com

**Rui Heng Yang**
Huawei Technologies Canada
rui.heng.yang@huawei.com

**Xuan Zhao**
Huawei Technologies Canada
xuan.zhao@h-partners.com

**Yasser H. Khalil**
Huawei Technologies Canada
yasser.khalil1@huawei.com

**Haoyu Lei**
Huawei
lei.haoyu@huawei.com

**Amir Rasouli**
Huawei Technologies Canada
amir.rasouli@huawei.com

**Yinchuan Li**[†]
Huawei
liyinchuan@huawei.com

## Abstract

Diffusion models, such as diffusion policy, have achieved state-of-the-art results in robotic manipulation by imitating expert demonstrations. While diffusion models were originally developed for vision tasks like image and video generation, many of their inference strategies have been directly transferred to control domains without adaptation. In this work, we show that by tailoring the denoising process to the specific characteristics of embodied AI tasks—particularly the structured, low-dimensional nature of action distributions—diffusion policies can operate effectively with as few as 5 neural function evaluations (NFE). Building on this insight, we propose a population-based sampling strategy, genetic denoising, which enhances both performance and stability by selecting denoising trajectories with low out-of-distribution risk. Our method solves challenging tasks with only 2 NFE while improving or matching performance. We evaluate our approach across 14 robotic manipulation tasks from D4RL and Robomimic, spanning multiple action horizons and inference budgets. In over 2 million evaluations, our method consistently outperforms standard diffusion-based policies, achieving up to 20% performance gains with significantly fewer inference steps.

## 1 Introduction

Stochastic policies have become increasingly important in robotic manipulation and more generally Embodied Artificial Intelligence (EAI), where agents must operate in real-world environments typically involving large action spaces, possibly stochastic responses, and multiple valid strategies for achieving the same objective [1]. These challenges are further compounded by data scarcity and

---

[*]Equal contributions
[†]Corresponding author

39th Conference on Neural Information Processing Systems (NeurIPS 2025).

the need for strong generalization from limited demonstrations. Diffusion-based policies [2, 3] offer a promising solution by learning to model the full distribution over expert actions, thereby mitigating mode collapse and enabling diverse robust behavior.

Despite their success, diffusion models suffer from a key drawback: inference is sequential and computationally expensive, requiring many denoising steps to produce high-quality samples. This latency is a major limitation for real-time applications in robotics, where fast and reliable action generation is critical. To address this, recent works have proposed distillation [4], consistency models [5, 6], and shortcut flow-matching [7], which trade off simplicity or performance for faster sampling by training new models.

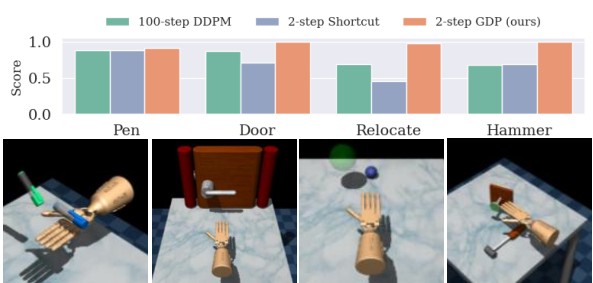

Figure 1: Comparison of normalized scores of Genetic Diffusion Policy to shortcut and diffusion policy baselines on Adroit Hand tasks.

In this work, we accelerate off-the-shelf diffusion policies without any retraining or architecture changes. We show that by reducing the number of inference steps and modifying the denoising schedule, we can often improve performance. Our analysis reveals that the default inference process suffers from out-of-distribution (OoD) intermediate states caused by clipping—a heuristic introduced to constrain predictions. Contrary to findings in image generation, we observe that reducing the injected noise in denoising steps improves performance in robotic tasks, due to the structured and low-dimensional nature of their action distributions. These findings emphasize a critical distinction: techniques that enhance image generation models do not necessarily transfer to embodied AI. Rather than blindly adopting heuristics from vision, robotic policy models require a dedicated analysis of their training dynamics, action spaces, and inference behavior.

To this end, we introduce the Genetic Diffusion Policy (GDP), see Figure 2, which uses a population-based selection mechanism to filter denoising trajectories based on an OoD score, reducing clipping artifacts and improving sample quality—especially at low step counts.

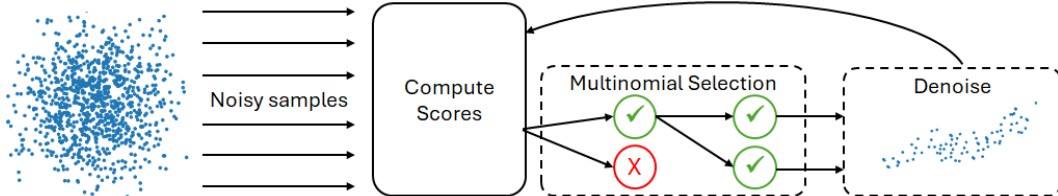

Figure 2: Genetic Denoising Process. One starts from pure Gaussian noisy samples, fitness scores are computed and used a weights for a multinomial selection. Selected samples are duplicated to replace deleted samples. Then, apply a denoising step following a possibly twisted DDPM denoising step. If not terminal denoising step loop back to computing fitness scores and repeat. By choosing a fitness score measuring whether samples are in distribution, the method favor more denoising trajectories with more precise denoising estimations hence more precise sampled actions.

We evaluate our method on 14 manipulation tasks from D4RL [8] and Robomimic [9], covering 6 action horizons and 18 inference budgets, using up to 500 seeds per configuration. Baselines include DDPM/DDIM [10, 11], EDM [12], and shortcut diffusion models [7]. Figure 1 summarizes our results on Adroit Hand tasks. In summary, our main contributions are:

- **Efficient Denoising:** We propose simple yet effective modifications to the denoising schedule that enable faster, more accurate sampling from existing diffusion policies.

- **Theoretical and Empirical Analysis:** We identify and explain the role of OoD noise and the counterintuitive benefits of reduced noise injection in low-dimensional robotic settings.

- **Genetic Diffusion Policy (GDP):** We introduce a novel sampling strategy that uses a population of candidate denoising trajectories and selects in-distribution paths to improve

robustness and performance at low inference budgets. To our knowledge, our method is the first attempt at employing genetic algorithms to accelerate diffusion model sampling.

## 2 Making better use of Diffusion Policies

### 2.1 Diffusion Policies

Robotic manipulation tasks are naturally modeled as Markov Decision Processes (MDPs), i.e. tuples $\langle \mathcal{S}, \mathcal{O}, \mathcal{A}, T, R, \pi \rangle$. Here $\mathcal{S}$ denotes the set of environment states, $\mathcal{O}$ the set of observations, $\mathcal{A}$ the action space, $T : \mathcal{S} \times \mathcal{A} \to \mathcal{S}$ the (possibly stochastic) transition kernel, and $R : \mathcal{S} \to \mathbb{R}_{\geq 0}$ a terminal-reward function. At time $t$ the physical world—including the robot—is in some $s_t \in \mathcal{S}$, while the agent only perceives a partial observation $o_t \in \mathcal{O}$ and selects an action $a_t \in \mathcal{A}$ according to a policy $\pi : \mathcal{O} \to \mathcal{A}$, after which the environment transitions to $s_{t+1} \sim T(s_t, a_t)$.

**Training.** Diffusion policies [2, 3] adapt the diffusion-model paradigm [10, 11] to decision making. Given a demonstration data set $\mathcal{D} = \{(o_t^{(i)}, a_t^{(i)})_{t=0}^{t^*}\}_{i=1}^{|\mathcal{D}|}$ comprising successful episodes, we construct a distribution
$$\mu\big(o_{t-h_{\mathcal{O}}:t+h_{\mathcal{A}}}, \ a_{t:t+h_{\mathcal{A}}}\big) \quad \text{on} \quad \mathcal{O}^{h_{\mathcal{O}}} \times \mathcal{A}^{h_{\mathcal{A}}},$$
i.e. windows of length $h_{\mathcal{O}}$ past observations and $h_{\mathcal{A}}$ future actions centered at some $\tau$. For numerical stability, we assume $\mathcal{A}$ is embedded in $\mathbb{R}^d$ and rescaled to $[-1, 1]^d$.

For a chosen noise schedule $(\alpha_t)_{t=0}^T$, the network $\epsilon_\theta : \mathcal{A}^{h_{\mathcal{A}}} \times \{0, \cdots, T\} \times \mathcal{O}^{h_{\mathcal{O}}} \to \mathcal{A}^{h_{\mathcal{A}}}$ is trained by minimizing the Denoising Diffusion Probabilistic Model (DDPM) loss

$$\mathcal{L}(\theta) = \mathbb{E}_{\substack{(o,x_0)\sim\mu \\ \epsilon\sim\mathcal{N}(0,\mathbf{I}) \\ t\sim\text{Unif}[T]}} \Big[ \| \epsilon_\theta(\sqrt{\overline{\alpha}_t}\, x_0 + \sqrt{1-\overline{\alpha}_t}\, \epsilon, \ t, o) - \epsilon \|_2^2 \Big] \quad \text{with} \quad \overline{\alpha}_t := \prod_{k=1}^t \alpha_k. \tag{1}$$

**Inference.** At test time we denoise an initial $x_T \sim \mathcal{N}(0, \mathbf{I})$ using

$$x_{t-1} = \sqrt{\overline{\alpha}_{t-1}} \operatorname{clip}\left[ \frac{x_t - \sqrt{1-\overline{\alpha}_t}\, \epsilon_\theta^{(t)}(x_t)}{\sqrt{\overline{\alpha}_t}} \right] + \sqrt{1-\overline{\alpha}_t - \sigma_t^2}\, \epsilon_\theta^{(t)}(x_t) + \gamma\, \sigma_t\, \epsilon_t, \tag{2}$$

where $\sigma_t^2 := \eta^2 \frac{(1-\overline{\alpha}_{t-1})(\overline{\alpha}_{t-1}-\alpha_t)}{\overline{\alpha}_{t-1}(1-\overline{\alpha}_t)}$, $\eta \in [0, 1]$ interpolates between the deterministic DDIM sampler ($\eta = 0$) and the stochastic DDPM sampler ($\eta = 1$), and $\gamma = 1$ in the standard formulation. Equation (2) is a finite-difference discretizations of the Stochastic Differential Equation (SDE) derived in [12],

$$dx_t = -\dot{\sigma}(t)\,\sigma(t)\, \nabla_x \log p(x_t \mid t, 0)\, dt \ - \beta(t)\sigma^2(t)\, \nabla_x \log p(x_t \mid t, o)\, dt + \gamma\sqrt{2\beta(t)}\,\sigma(t)\, dW_t, \tag{3}$$

coupling the probability-flow ODE with a Langevin diffusion. Separating the training horizon $T$ from a possibly reduced number of inference steps $\delta$ is straightforward: replace $(t, t-1)$ in (2) and in $\sigma_t$ by $(t_j, t_{j-1})$ for a monotone schedule $0 \leq t_0 < \cdots < t_\delta \leq T$.

### 2.2 Clipping-Induced Denoising Defects

The `clip` operation in (2) serves two independent purposes. First, robotic actions are normalized to $[-1, 1]^d$, so any estimator $\hat{x}_0$ lying outside that cube must be projected back. Second, for the first few timesteps of a cosine noise schedule [13] we have $\overline{\alpha}_t \approx 0$, making the denominator $\sqrt{\overline{\alpha}_t}$ ill-conditioned; clipping prevents numerical explosions.

Unfortunately, this crude safeguard creates a subtle distributional mismatch. Near $t = T$ most coordinates of $\hat{x}_0$ saturate at $\{-1, 1\}$, so the inference distribution $x_t \sim \frac{1}{\sigma}\big(\mathcal{N}(0, \sigma^2\mathbf{I}) \otimes (\hat{\mu} \mid o)\big)$ is supported almost exclusively on the corners of the hyper-cube, whereas training sees $\mathcal{N}(0, \sigma^2\mathbf{I}) \otimes (\mu \mid o)$, whose conditional $\mu$ is spread throughout the interior. Consequently,

1. early denoising steps convey little task-relevant signal;
2. the predictor $\epsilon_\theta$ incurs larger errors (because it never learned to handle these extreme inputs);
3. the overall reverse process wastes iterations before trajectories re-enter the high-density region of the noised target action manifold.

Figure 3 quantifies this phenomenon across tasks and sampler hyper-parameters: the higher the clipping frequency, the lower the final episodic return. Similar training–sampling discrepancies were found to degrade quality in other domains and have motivated methods such as InferGrad [14], which explicitly harmonize the two regimes.

## 2.3   Exploration-Exploitation trade-off in Diffusion Policy for Robotics

The link between clipping frequency and score suggests that significant improvement may be achieved by training a diffusion model well so that it generates a good policy, possibly with many denoising steps, then tweaking the denoising process to obtain sufficient sample quality with less denoising steps. We argue that the parameters $\eta$ and $\gamma$ allow an exploration-exploitation trade-off that one may leverage to mitigate the clipping issue. As discussed in the introduction, generalization requires to train a stochastic policy to mitigate mode collapse during training. However, mode collapse is not an issue during inference: a robot that always chooses the same solution given the same context is acceptable as long as it is successful. In other words, if the diffusion model is well trained and fits the whole training distribution, exploitation in the sense of outputting less diverse but acceptable actions should not be an issue.

On the one hand, looking at Equation 3, by reducing the noise injection scale $\gamma$, our denoising process collapses to the deterministic probability flow (which is a gradient ascent of the noise target distribution) with noise decay. The smaller the $\gamma$, the more likely the generated sample lies close to the maximum of density of a dominant mode in the target action distribution. We favor exploitation over exploration.

On the other hand, lack of exploration may result in missing the best modes. Reduction of noise injection and number of denoising steps results in a bias toward modes closer to 0. In the context of image generation it results in caricatural outputs: as depicted in Figure 4, people portrait generation yields monstrous low contrast faces with protruding eyes. Few-steps with little noise injection is putting a strain on the ability of the denoising process to find good modes.

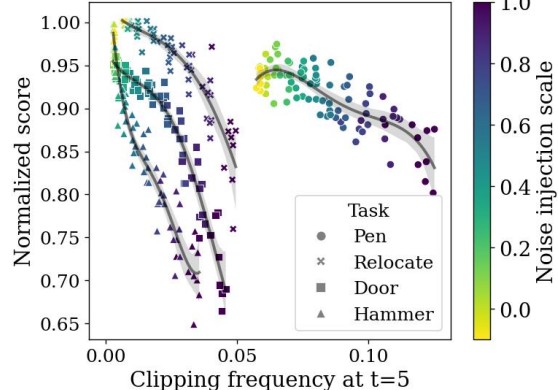

Figure 3: Each marker represents a unique combination of task, denoising-step count, and noise-injection scale, all at denoising step $t = 5$. Clipping frequency is calculated as the proportion of entries in the flattened, noised action-sequence tensor that are projected onto the cube boundary. Grey lines show fourth-order polynomial regressions fitted separately for each task.

Furthermore, robotic manipulation distributions conditioned by observations are intrinsically low dimensional and simpler than image distributions: the image dataset CelebA has extrinsic dimension $2^{16}$, intrinsic dimension around 25 [15] while the action space of Adroit Hand environment varies from 24 to 30 with an estimated intrinsic dimension 11 at action horizon 24 for most tasks, see Appendix B. Also, Robotic tasks being MDP and assuming the action horizon is small enough, mistakes may be corrected as long as they are not fatal. This translates into robustness against imperfect policies.

## 2.4   Simple Empirical Solutions

The discussion of the previous section suggests simple ways to improve our usage of the Exploration-Exploitation tradeoff. First, it is likely that a well-chosen denoising time schedule $(t_j)_{j \in \{1, \cdots, \delta\}}$ allows to reduce the number of needed steps. Starting the denoising process from $t_\delta < T$ eliminates an uninformative step, the normal distribution from which $x_{t_\delta}$ is drawn already covers the noised target distribution. Also, less steps means less injected noise, less clipping and less OoD values hence better quality of noise prediction $\epsilon_\theta(x_t; t, o)$.

Second, reduce the noise injection scale favoring exploitation, reducing the probability of denoising into tensor of larger values hence larger clipping frequency.

Third, use a clipping-free denoising process. One may replace the DDPM scheduler with a DDIM scheduler. In Chi et al. [3], DDIM was used as way to do less denoising steps transposing the initial purpose of Song et al. [11], however they do not benchmark DDIM in emulated environments. Our experiments found that DDIM do not perform particularly well. This is not fortuitous: DDIM compensates the lower noise with larger deterministic steps, which are unreliable at the beginning of the denoising process due to OoD partially denoised samples. A more principled training and sampling that we denote by EDM was proposed by Karras et al. [12]. In principle, one could use an off-the-shelf DDPM Diffusion Policy model and use an EDM sampler. However, in order to obtain competitive results, not only we had to retrain an EDM policy from scratch but also had to tune training hyperparameters (see Experiments section 4). Unlike

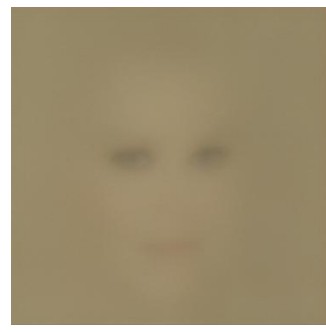

Figure 4: Diffusion-generated face without noise injection.

DDPM, the EDM policies were very unreliable across horizons, going from solving an environement in horzion 24 to not even reaching 50% success rate as shown in figure.

The first two solutions above are easy to implement and yield significative improvement over baseline as the Experiments section 4 suggests. However, further improvement seems rather limited as these methods do not give much room to improve the aforementioned trade-off.

## 3   Genetic Diffusion Policy

The Exploration-Exploitation trade-off discussed in the last section is solved in diffusion for image generation by long denoising trajectories. However, in terms of complexity constraints, EAI is a polar opposite of Image generation: image generation is limited in the memory complexity of algorithms because of the high dimension of the distributions, but time is not much of an issue image generation tasks rarely require very high reactivity. On the other hand, EAI requires fast generation, but is less constrained memory-wise thanks to the low dimensionality of the action space. Our solution to improve both exploration and exploitation is to leverage this specificity of EAI by enhancing the denoising process using a genetic algorithm. The heuristics we choose for our genetic algorithm measures how OoD a given sample is.

---

**Algorithm 1** Genetic Diffusion Policy

---

**Require:** Diffusion Policy noise model $\epsilon_\theta$ with schedule $(\alpha_t)_{t \in [0,T]}$. Stochastic denoising rule $x_{t_{j-1}} = D(x_{t_j}, j, \epsilon_\theta(x_{t_j}^i, t_j))$. OoD score $\varphi(x_{t_j}^i, t_j, \epsilon_\theta(x_{t_j}^i, t_j))$. Population size $P$. Survival number $S$. Denoising steps $N$.
1: Sample $x_{t_N}^i \sim \mathcal{N}(0, 1)$ for $i \in \{1, \cdots, P\}$
2: $j \leftarrow N$
3: **while** $j \neq 0$ **do**
4:      $j \leftarrow j - 1$
5:      Compute scores $p_i := \varphi(x_{t_j}^i, t_j, \epsilon_\theta(x_{t_j}^i, t_j))$
6:      Select $S$ element in $\{1, \cdots, P\}$ with $(i_1, \cdots, i_S) \sim \text{Multinomial}(S, p_1, \cdots, p_P)$
7:      $x_{t_{j-1}}^i \leftarrow D(x_{t_j}^{i_{i\%S}}, j, \epsilon_\theta(x_{t_j}^{i_{i\%S}}, t_j))$
8: **end while**
9: Return $x_0^0$

---

In details, we generate a population of noised samples; Before applying a denoising step, we compute a fitness score for each partially denoised sample, then select half of the sample using a Multinomial sampler weighted by the fitness scores, and finally duplicate the selected samples to fill the population batch. Then a usual denoising step is applied to the population. Our fitness score is chosen to enhance our diffusion models by favoring in distribution samples. Two families of scores $\varphi$ are considered:

$$\varphi_{\text{stein}, f, T}(x_t, t) = T \times f(\|\epsilon_\theta(x_t, t)\|) \quad \text{and} \quad \varphi_{\text{clip}, f, T}(x_t, t) = T \times f(\hat{x}_0 - \text{CLIP}(\hat{x}_0)), \quad (4)$$

where $T$ is a temperature factor, $f$ is a scaling function and $\hat{x}_0 := \frac{x_t - \sqrt{1 - \overline{\alpha}_t} \epsilon_\theta(x_t, t)}{\sqrt{\overline{\alpha}_t}}$. See Algorithm 1.

The clip-based score family $\varphi_{\text{clip}, f, T}$ is clearly motivated by the discussion of section 2.2 both theoretically and empirically. Theoretically, clipping occurs when $\hat{x}_0$ is OoD, hence the importance

Table 1: **Adroit Hand results.** Normalized success rates averaged over 100 seeds. DDIM and ablation variants are integrated. "Schedule" refers to adapted schedule; "Best $\gamma$" refers to best reduced noise scale ($\gamma$=0.2).

| Method | Steps | $\gamma$ | Hammer | Relocate | Pen | Door |
|---|---|---|---|---|---|---|
| *Full diffusion schedule (100 steps)* | | | | | | |
| DDPM | 100 | 1 | 0.68 | 0.69 | 0.88 | 0.87 |
| DDPM | 100 | 0 | **0.99** | *0.95* | **0.94** | **1.00** |
| Shortcut | 100 | – | 0.70 | 0.84 | 0.81 | 0.87 |
| DDIM | 100 | – | 0.70 | 0.38 | 0.50 | 0.83 |
| GDP | 100 | 0.2 | **0.99** | **0.98** | **0.94** | **1.00** |
| *Few-step inference (5 steps)* | | | | | | |
| DDPM | 5 | 1 | 0.91 | 0.91 | 0.85 | **1.00** |
| DDPM | 5 | 0 | *0.99* | 0.97 | *0.84* | **1.00** |
| Shortcut | 5 | – | 0.88 | **1.00** | 0.81 | 0.94 |
| DDIM | 5 | – | 0.71 | 0.38 | 0.70 | 0.81 |
| GDP | 5 | 0.2 | **1.00** | *0.99* | **0.91** | **1.00** |
| *Minimal inference (2 steps)* | | | | | | |
| DDPM | 2 | 1 | 0.00 | 0.01 | 0.13 | 0.01 |
| DDPM | 2 | 0 | 0.00 | 0.02 | 0.11 | 0.01 |
| Shortcut | 2 | – | 0.88 | **1.00** | 0.81 | 0.94 |
| DDIM | 2 | – | 0.76 | 0.42 | 0.73 | 0.95 |
| DDPM + Schedule | 2 | 1 | 0.87 | 0.64 | 0.74 | 0.97 |
| DDPM + Schedule | 2 | 0 | 0.95 | 0.74 | 0.75 | **1.00** |
| DDPM + Schedule + Best $\gamma$ | 2 | 0.2 | *0.98* | 0.92 | *0.89* | **1.00** |
| GDP | 2 | 0.2 | **1.00** | *0.98* | **0.91** | **1.00** |
| Shortcut | 1 | – | 0.83 | 0.93 | 0.74 | 0.89 |

of clipped coordinates is a measure of OoD. Empirically, if clipping is the issue, favoring less clipped samples should yield significant improvement in sample quality.

The stein-based score $\varphi_{\text{stein},f,T}$ has a double theoretical motivation. First, reducing the noise injection breaks the Langevin process, a stein-fitness emulates the noise injection of the Langevin process. Second, the noise estimator is a direct measure of OoD since a high noise means that the sample is far away from every modes of the target distribution.

## 4 Experiments

### 4.1 Experimental Setup

We evaluate our hypotheses that (*i*) reducing the number of inference iterations and (*ii*) lowering the noise injection scale both mitigate clipping and improve performance. Experiments are conducted on the **Adroit Hand** [16], see figure 1, and **Robomimic** [9] benchmarks. Each Adroit task involves controlling a 24–30 DoF robotic hand to accomplish a distinct goal:

- **Pen:** Orient a pen to a target angle.
- **Relocate:** Grasp a ball and place it at a goal position.
- **Hammer:** Pick up a hammer and strike a nail.
- **Door:** Pull down the latch and open the door.

All methods use the same **UNet** architecture with 65M parameters from the official Diffusion Policy (DP) implementation [17] to ensure fairness. Each Adroit configuration is evaluated on 100 seeds, and each Robomimic configuration on 500 seeds.

We sweep over the following inference hyperparameters:

- Number of inference **steps** $\delta \in \{1, 2, \ldots, 10\} \cup \{20, 30, \ldots, 100\}$,

- **Noise scaling** factor $\gamma \in \{0.0, 0.1, \ldots, 1.0\}$,
- **Action horizon** $h_{\mathcal{A}} \in \{24, 48, 76, 100, 152, 200\}$,
- Sampling **method**: DDPM, GDP, DDIM or Shortcut.

Since publicly released Adroit checkpoints do not cover all action horizons, we retrained each diffusion policy using the DP pipeline with AdamW [18, 19], learning rate $10^{-4}$, weight decay $10^{-6}$, batch size 64, and 200 epochs. Shortcut models [7] were re-implemented in PyTorch and trained via 10 random hyperparameter seeds per task–horizon pair, keeping the best model. All base models were trained for 100 diffusion steps (128 for Shortcut) and evaluated with a subsampled cosine inference schedule. A linear schedule was also tested but underperformed consistently across all methods. The same DDPM-trained checkpoint is used for DDPM, GDP and DDIM, while the shortcut model is trained separately.

We then test the proposed Genetic algorithm with a very coarse parameter grid as the goal is to justify the use of the algorithm 1, rather than tuning it to its most optimal state. We test populations $p \in [4, 8, 16, 32]$, temperatures $t \in [1, 10, 100, 1000]$, and noise scales $\gamma \in [1, 0.6, 0.3, 0.2, 0.1]$.

## 4.2 Results

**Number of diffusion steps -** The correlation between the number of steps $n$ and clipping is evident across all tasks: reducing the number of diffusion steps lowers the amount of clipping. This leads to similar or better scores with a greatly reduced inference cost. We observe a peak, prior to $n = 4$, at which the clip is minimal, i.e. the performance is optimal, see figure 5(a). The location of the peak is slightly task dependent but robust to different seeds. This allows one to tune the model to the optimal number of steps, without it breaking in unfamiliar situations.

**Noise injection scale** The reduced noise inference processes also behave as expected: lowering the injected noise reduces clipping significantly. However, as shown in Figure 3, the extent to which the clipping can be mitigated via noise reduction is limited. This means that a part of the clipping is inherent to the deterministic part of the denoising process, and not only to the noise injection from the Langevin process. Still, the increase in score resulting from using lower noise scales is significant, see figure 5(b). With this noise rescaling, the same UNet can go from a 75% success rate to totally solving the task. This also stabilizes the score in higher numbers of denoising step, bridging the gap between the peak and the rest of the distribution.

**Horizon** Throughout all our experiments, all the behaviors mentioned in the previous paragraphs are present across horizons. We notice that intermediate horizons seem to be harder than either short, or full length horizons. Our hypothesis for this phenomenon is that by increasing the horizon, we tradeoff conditioning complexity for distribution complexity. In longer horizons, the model has to learn a few complex distributions whereas in shorter horizons, the distributions are simpler but the conditioning needs to encapsulate more of the dynamics of the environment. This would lead to a dip where the tradeoff is suboptimal, with a significant number of non trivial distributions to learn. This tendency was even more present with EDM, as showed in Figure 5(c)

**Shortcut Models** In this experiment, we evaluate our approach against the shortcut models. Shortcut models [7], as prominent means of increasing sampling speed, train a model with a conditioning on the number of steps that will be taken in total, which enables outputting quality samples in one step. However, this speedup comes at the price of performance degradation. As shown in Frans et al.[7], DP's success rate can be reduced by as much as 20% on tasks such as the robomimic transport[9] task. We train and tune shortcut models for all four Adroit tasks, conducting 10 training trials for each. The reported performance for the shortcut method on each environment is the highest score achieved across the 10 trained models.

The results in Table 1 demonstrate that although the shortcut method is the only one capable of achieving 1-step sampling, its performance falls significantly short. Our approach to 2-step generation combines the GDP technique with the timestep adjustments described in Section 2.4, where we set the maximum timestep to 90 and the minimum to 20. This method outperforms baselines or is at least on par —regardless of the number of diffusion steps used—while requiring only two steps. Starting from a solid off-the-shelf model with approximately a 70% success rate, our method achieves a 100%

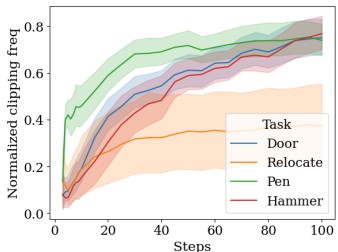
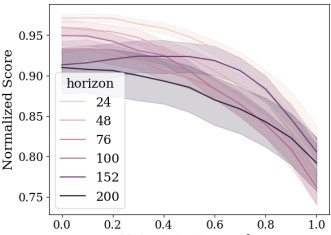
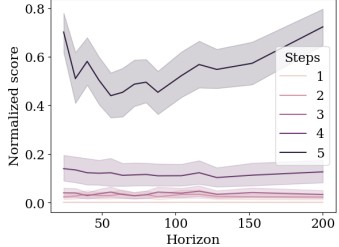

(a): Clip frequency vs. denoising steps (Adroit).

(b): Effect of noise injection across horizons.

(c): EDM: score vs. horizon (few-step).

Figure 5: (a) Clip frequency normalized by the maximum observed value as a function of the number of denoising steps; clipping increases with more steps. (b) Impact of noise injection scale across action horizons, averaged over tasks; lower injection improves performance systematically. (c) Normalized score across horizons for tuned EDM in the few-step regime (for $n > 5$, curves match $n = 5$).

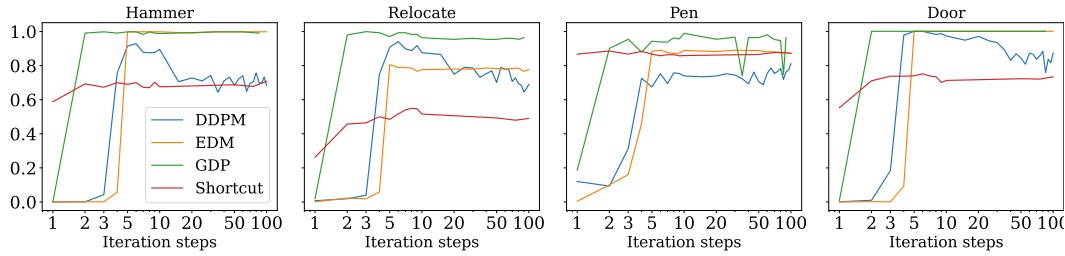

Figure 6: Performance across timesteps for all tasks.

success rate along with a substantial speedup. Our experiments also showed that GDP suffers from using $\gamma = 1$. We posit that this is due to the noise enabling individuals from the populations to "jump" from mode to mode even at later steps, causing mode collapse due to the survivor selection process.

### 4.3 Inference Overhead and Wall-Clock Runtime

We measure step-wise overhead on an RTX 3080, batching the population in a single forward pass per step. The *NFE cost* is the wall-time of a single call of the model on the population; the *Step cost* is the wall-time of the denoising step after NFE: computation of formula 2, computation of fitness score and population management. See table 3.

## 5 Related Works

First, the Shortcut baseline maybe seen as a self-distillated consistency model [5, 4]. More traditional knowledge distillation methods maybe employed to accelerate diffusion models [20].

Second, fast generation of diffusion models is a very active research thread mostly trying to improve the Stochastic Differential Equation (SDE) solver using in the denoising process [21, 12, 22] while other approaches try to leverage a parallel sampling [23]. This last method may be compared to ours as their averaging method may be interpreted as a cross-breeding step. More generally, swarm methods for solving Partial Differential Equations (PDE) is active research thread [24]. Via Fokker-Planck equations, the PDE and SDE view points are dual to one another.

Third, by leveraging a simple metaheuristic on sample population, our Genetic Denoising Process open the door for a broad family of metaheuristics [25]. Our stein fitness score may be related to PDE-constrained swarm optimization [26].

# 6 Limitations

First, most adaptations to the diffusion process made in this paper are only valid in the context of EAI tasks. Indeed, genetic algorithms would not be suitable for image generation as they drastically increase memory costs which are already an issue for high resolution image datasets. The tweaks analysed in Section 2.4 are also not as viable for image generation : Song et al. [11] show that reducing the number of inference steps largely decreases the performance of the models, and contrary to what we observe in our tasks, DDIM is actually the better option for fast image sampling. As already discussed in section 2.3, reducing the noise injection scales leads to unsatisfactory results for image generations.

Second, our genetic algorithm is the simplest possible since it includes neither cross-breeding nor sophisticated mutations. Our metaheuristic is very simple and purely local. For instance, diversity control of swarm optimization [27, 28] may be employed. Mutation and cross-breeding taken from image (non-diffusion) denoising may also be considered [29].

Third, there is a clear difference between robomimic and Adroit Hand tasks. Since the extrinsic dimension of the action space of the former is 3 times smaller than that of the latter, we hypothesize that robomimic tasks target distributions are significantly simpler than that of Adroit Hand. However, each robomimic task comprises more various manipulations suggesting a more diverse conditioning. As a result, little improvement may be achieved by improving the denoising process, the bottleneck is expected to be the conditioning of the noise model $\epsilon_\theta$. We did not test this hypothesis and our method cannot solve this problem.

Finally, our theoretical analysis remains preliminary. We conjecture that the expected denoising error can be bounded by an increasing function of the expected Stein score norm computed along the denoising trajectory. Moreover, a non-rigorous derivation suggests that, in the limit of an infinite population and infinitesimal step size, multinomial population denoising approximates the addition of a stochastic noise term together with a gradient-ascent term on the fitness score. To the best of our knowledge, the effect of noise-scale manipulation has not yet been studied from a theoretical standpoint. It is reasonable to expect that varying the noise scale preserves the support of the learned distribution while inducing a bias toward its mean when the noise is reduced. Formal statements and proofs of these relationships are left for future work.

# 7 Conclusion

All in all our experiments demonstrate that using our proposed approach, we can improve performance as well as sampling rate of off-the-shelf-models. Additionally, using Genetic Denoising can help further improve model accuracy and stability, even with simple estimators of sample quality. We showed that off-the-shelf models can be used for two-steps inference with better performance compared shortcut models. We conclude that it is possible to exploit this type of inference framework to extract even more performance out of a given model provided the model is well trained and the target distribution sophisticated enough.

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

# A   Extra Experimental Details

## A.1   Image generation

In the Simple Empirical Solutions section, we show an image sample illustrating a mode collapse. This image was obtained using the **google/ddpm-celebahq-256** pretrained pipeline [30]. We tweak the associated scheduler to use $\gamma = 0$, and run inference while setting the number of inference steps to 50. Note that with the initial number of steps, the sample converges to a uniformly gray square.

# B   Intrinsic Distribution Manifold Dimension Estimation

We subscribe to the Manifold Hypothesis [31, 32] stating that data distributions are supported by a submanifold of $\mathbb{R}^n$. The intrinsic dimension of a dataset refers to the minimum dimension of such a manifolds supporting the whole dataset. Several notions of intrinsic dimension of a dataset as usually considered, the most common are based on Minkoswki or Hausdorff dimensions [33, 34, 35, 15]. Since our work focuses on diffusion models, we favor the method proposed by Stanczuk et al. [36]. This methods allows to estimate the intrinsic dimension learned distribution of a diffusion model by doing a PCA of different values of the noise model $\epsilon_\theta$.

# C   Extra Experimental Results

## C.1   Link between clipping and Score

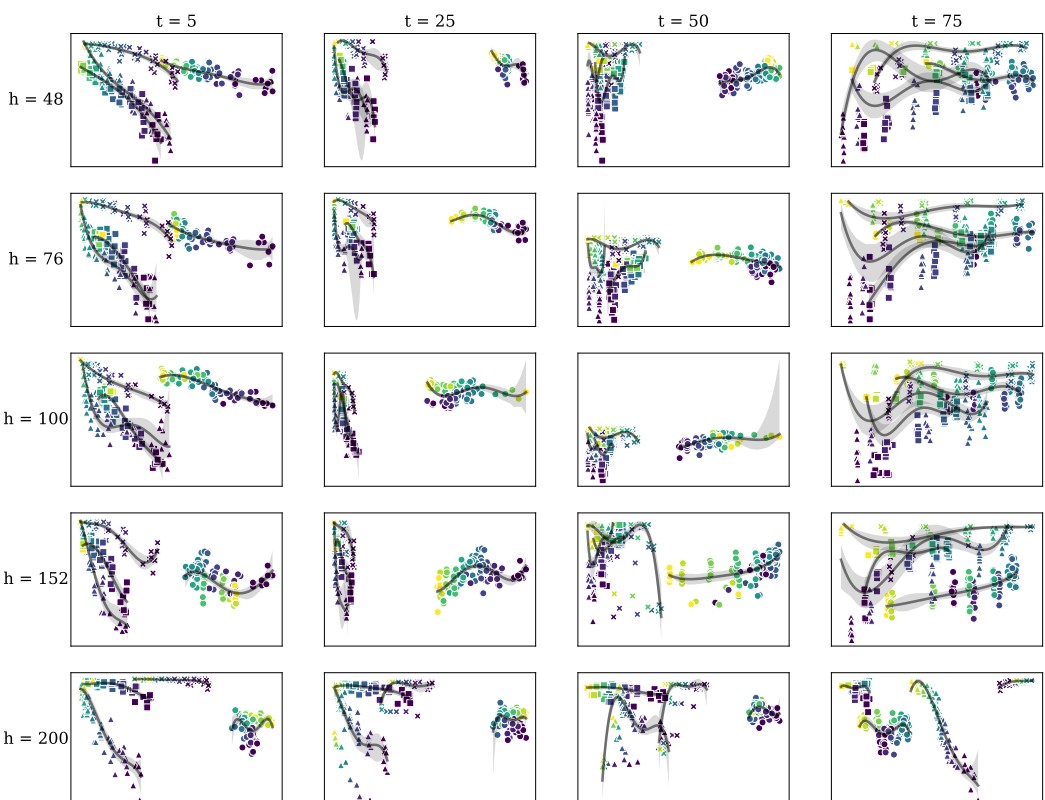

Figure 7: Same as Figure 3, but across action horizons $h$ and timesteps $t$. Hue and marker styles are also preserved.

## C.2 DDIM

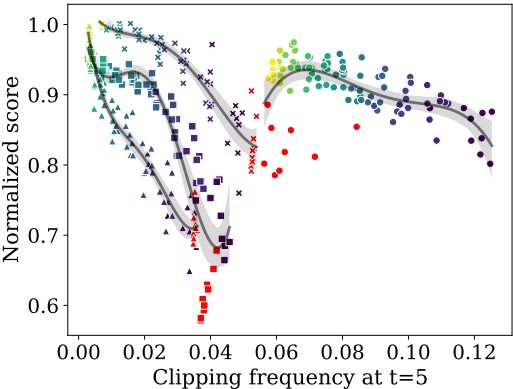

Figure 8: Same as Figure 3. Hue and marker styles are also preserved, with the added DDIM data points in red

Table 2: **Robomimic results.** Normalized success rates over 500 seeds. PH and MH denote the training dataset used for training: Proficient Human and Medium performing human. "DDPM+Schedule." refers to adapted schedule variant of DDPM. Notice that most success rates are within 2 standard deviations of GDP suggesting that GDP fails to improve the base policy over adapted schedule. We hypothesise that the quality of the base model is a bottleneck as similar unclear results where observed on Adroit when using less potent checkpoints.

| Method | $\gamma$ | Lift PH | Lift MH | Can PH | Can MH | Square PH | Square MH | Transport PH | Transport MH | ToolHang PH |
|---|---|---|---|---|---|---|---|---|---|---|
| *100-step inference* | | | | | | | | | | |
| DDPM | 1 | 1.00 | 1.00 | 0.97 | 0.95 | 0.92 | 0.85 | 0.84 | 0.62 | 0.53 |
| DDPM | 0 | 1.00 | 1.00 | 0.99 | 0.96 | 0.92 | 0.86 | 0.84 | 0.60 | 0.53 |
| GDP | 0.2 | 1.00 | 1.00 | 1.00 | 1.00 | 0.90 | 0.86 | 0.84 | 0.64 | 0.53 |
| DDIM | – | 0.998 | 0.998 | 0.982 | 0.970 | 0.928 | 0.846 | 0.846 | 0.606 | 0.514 |
| *5-step inference* | | | | | | | | | | |
| DDPM | 1 | 1.00 | 1.00 | 1.00 | 0.96 | 0.94 | 0.85 | 0.81 | 0.58 | 0.55 |
| DDPM | 0 | 1.00 | 0.99 | 0.99 | 0.97 | 0.92 | 0.85 | 0.83 | 0.61 | 0.52 |
| GDP | 0.2 | 1.00 | 1.00 | 0.99 | 0.97 | 0.95 | 0.86 | 0.77 | 0.59 | 0.50 |
| DDIM | – | 0.998 | 0.998 | 0.954 | 0.956 | 0.898 | 0.826 | 0.808 | 0.606 | 0.510 |
| *2-step inference* | | | | | | | | | | |
| GDP | 0.2 | 1.00 | 1.00 | 0.99 | 0.97 | 0.92 | 0.84 | 0.77 | 0.58 | 0.49 |
| DDPM + Schedule | 0 | 0.998 | 0.995 | 0.989 | 0.968 | 0.919 | 0.826 | 0.786 | 0.595 | 0.481 |
| DDPM + Schedule | 1 | 1 | 0.998 | 0.984 | 0.970 | 0.934 | 0.845 | 0.818 | 0.578 | 0.502 |
| DDIM | – | 0.998 | 1.000 | 0.982 | 0.966 | 0.922 | 0.842 | 0.821 | 0.604 | 0.480 |

| Population | NFE cost ($\mu$s) | Step cost ($\mu$s) | Overhead ratio | Notes |
|---|---|---|---|---|
| 1 (DDPM) | 3800 | 200 | 1.00 | baseline |
| 8 | 3800 | 500 | 1.08 | under-utilized GPU |
| 16 | 4000 | 800 | 1.20 | |
| 32 | 4500 | 1500 | 1.50 | |
| 64 | 5500 | 2400 | 1.98 | memory-limited |

Table 3: Inference wall-clock time comparison. The overhead ratio is the ratios of sum of NFE and step costs between population 1-64 (GDP) and population 1 (DDPM).

**Supplementary Materials**

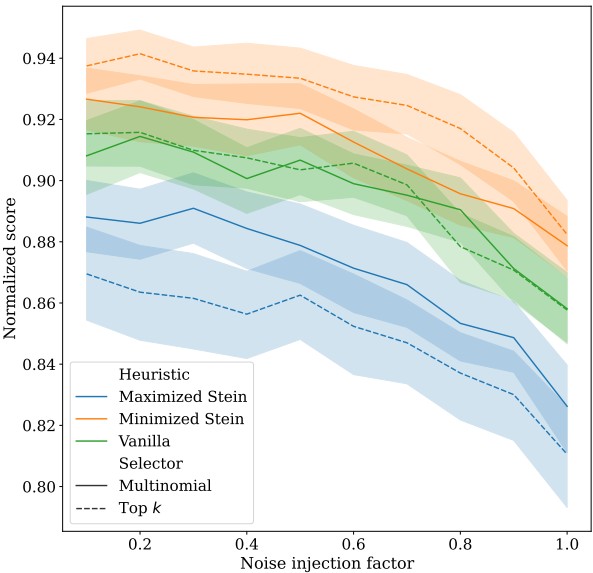

Figure 9: Normalized score across all noise injection factors, for 2-step diffusion. Results averaged over all Adroit tasks, with 20 trials of 100 environments each. The score function is used as the genetic algorithm heuristic. The selector picks using the given heuristic. *Multinomial* uses a temperature of 1, *top k* takes the best samples in a sorted order - the highest score sample is selected at $t = 0$. We use a population of 16 on all runs.

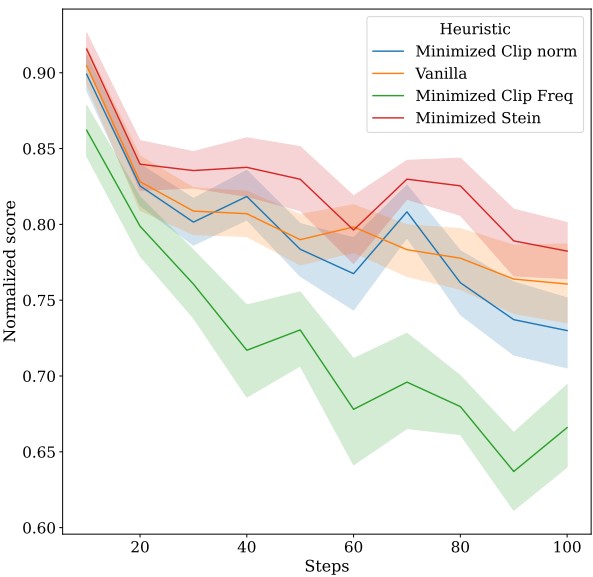

Figure 10: Normalized score across all numbers of steps, for $\gamma = 1$, given different genetic algorithm selection heuristics. Results averaged over all Adroit tasks, with 20 trials of 100 environments each. We use a population of 16 on all runs.

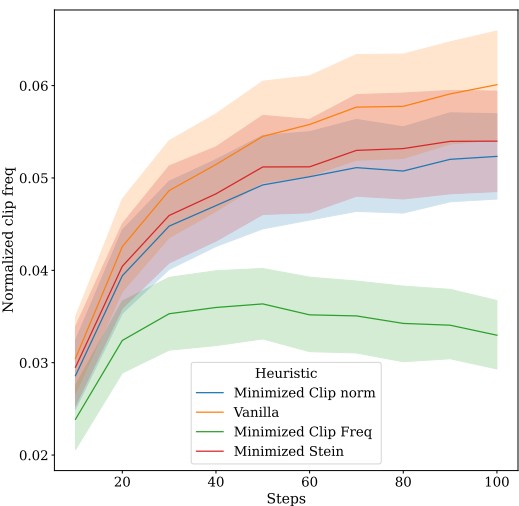

Figure 11: Normalized clipping frequency across all numbers of steps, for $\gamma = 1$, given different genetic algorithm selection heuristics. Results averaged over all Adroit tasks, with 20 trials of 100 environments each. We use a population of 16 on all runs.

These figures show that minimizing the stein score (or the norm of the estimated noise) is the best of simple genetic algorithm heuristics. Using clipping statistics as a heuristics distorts the sampled distribution by removing the mode with large values, when the stein based heuristic only measures how out of distribution the current intermediary sample is.

**Videos** All *gdp* videos are generated using a genetic algorithm using $\gamma = 0.2$, with stein score as heuristic. All non *gdp* videos videos are generated using a vanilla policy using $\gamma = 1$. For the 2-step diffusion, we use 80 as the maximum timestep and 20 as the minimum timestep.

