# OpenReview forum: "Two-Steps Diffusion Policy for Robotic Manipulation via Genetic Denoising"
_NeurIPS.cc/2025/Conference — NeurIPS 2025 poster_

### Official Review · Reviewer_ptGF · 2025-06-28

**Clarity:** 3
**Significance:** 2
**Originality:** 3
**Rating:** 4
**Confidence:** 4

**Summary:**

This paper presents an inference-time acceleration method for diffusion policies in robotic manipulation, leveraging a population-based strategy termed Genetic Denoising. The key insight is that the standard diffusion denoising process is often inefficient and overkill for low-dimensional action spaces in control tasks. The authors propose reducing the number of neural evaluations during inference to as few as two. They introduce a selection-based denoising scheme, where candidate trajectories are scored and evolved to minimize out-of-distribution risk. The method is evaluated across 14 D4RL and Robomimic tasks, showing improved or comparable performance to baselines, while significantly reducing compute.

**Questions:**

- What's the actual inference runtime for the method? Does the trajectory selection (scoring, sorting) process introduce overhead? What are the runtime comparisons with other baselines?
- Can the method scale to tasks with higher-dimensional actions (e.g., 30-DoF humanoid or dexterous hand control)? How does the performance degrade with action space size?

**Ethical Concerns:**

["NO or VERY MINOR ethics concerns only"]

**Final Justification:**

My concerns have been addressed during rebuttal and I lean towards accepting the paper.

**Limitations:**

Yes

**Quality:**

2

**Strengths And Weaknesses:**

Strengths
- The paper proposes a novel genetic denoising approach - using a population-based search to select good candidates based on ood scores. This approach allows using much fewer denoising steps.
- The paper highlights that in low-dimensional action spaces, the final denoised sample is often recoverable from early steps. This motivates the reduced-step formulation.
- The method does not require retraining the diffusion policy, making it a practical and cheap improvement to existing models.

Weakness
- Despite extensive simulation results, there are no real-world robot experiments to verify that the proposed fast denoising retains robustness when deployed on hardware. Real-world experiments could also show the benefits of faster inference by allowing dynamic manipulation tasks.
- The paper argues that mode collapse is not a problem in robotics tasks. However, multi-modal action distributions can be valuable, especially for ambiguous goals or multi-task policies. The value of sampling diversity at inference time (e.g., for steering or preference selection) is neglected.
- More analysis and theoretical justification on when the method is robust under heavy step reduction and when the method might fail (e.g., under high stochasticity or noisy dynamics) would be helpful.

---

> ### Author Rebuttal · Authors · 2025-07-31
>
> We thank the reviewer for their feedback.  We are pleased that they find our method well-motivated and practical. We respond to their comments and questions below.
>
> ## 1. Inference runtime
> (same answer as to reviewer fdQZ)
> We measured the step-wise overhead for GDP using a standard RTX 3080 GPU without specific optimization beyond calling the model batch-wise on the population.
>
> |Pop. Size |NFE cost ($\mu$s)| Step cost  ($\mu$s)|  Overhead ratio|
> |:----:|:----:|:----:|:----:|
> |1 (DDPM)|3800|200|1|
> |8|3800|500|1.075|
> |16|4000|800|1.2|
> |32|4500|1500|1.5|b
> |64|5500|2400|1.975|
>
> The relatively low cost is due to the under-utilization of the GPU computation and memory abilities. As the model grows bigger, the population may be more constrained. We argue the following.
> - In production, the observation embedding may represent a significant portion of the computational budget but since it can be shared across the population, its impact on the overhead is limited.
> - Training is typically done with a batch size larger than one; thus, assuming training and inference are done on the same hardware, inference is likely to leave room for a non-trivial population.
> - Fully-fused models, weight quantization, and other low-level optimization would allow for a larger batch size (hence a population) during inference.
>
> These points will be discussed in a dedicated section in the appendix.
>
> ## 2 . Method Scalability
>
> Our experiments include Robomimic and Adroit, which span low to medium action dimensionality, 9 and up to 30 DoF respectively. Results indicate that the method scales well and tends to be more beneficial in complex tasks, particularly those with more convoluted action distributions,  even at higher prediction horizons. As noted in answer to reviewer fdQZ, despite the low action space dimensionality, our experiments with long horizons show the method is efficient for extrinsic distribution dimension comparable to CIFAR or ImageNet-64.
>
> We have not tested the method on humanoid-level tasks (~60 DoF). Still, the demonstrated scalability up to 30 DoF and 200-step horizons suggests that the method would extend to such domains, at least for short horizons.
>
> ## 3. Real world experiment
> We tested our method on a pick-and-place task using a Franka Emika Panda arm. All methods—DDPM with 100 steps, 5 steps, 2 steps (with adapted schedules and zero variance), and GDP—achieved a 100% success rate. Although this does not highlight performance differences, it demonstrates that our approach is robust and deployable on real hardware for such tasks. Results and experimental details will be included in the paper.
>
> However, showcasing dynamic benefits from faster inference in real-world manipulation remains impractical for now. Such demonstrations would require a robotic platform capable of ultra-fast closed-loop control and streamlined integration between perception, model inference, and actuation, which are beyond our current resources.
>
> Nevertheless, we argue that reducing inference cost provides real benefits:
> - Enabling more reactive or frequent control cycles;
> - Freeing GPU compute for parallel tasks (e.g., perception or planning);
> - Allowing deeper models.
> - Allowing deployment on less expensive hardware.
>
>
> ## 4. Mode Collapse
>
> We agree that mode collapse may harm sample diversity, particularly in guided generation. We also acknowledge that guidance effectiveness typically declines as the number of denoising steps is reduced.
>
> Our proposed mechanism enables a diversity trade-off: possibly lowering the diversity with $ \gamma $ while compensating with a larger population. It is important to note that lowering $\gamma$ distorts the generated distribution, but it does not necessarily reduce mode coverage, just like increasing it does not necessarily increase mode coverage.
>
> In future work, we intend to explore a more integrated use of guidance with genetic denoising. For instance, applying guidance before selection and then scoring candidates with a Stein metric may preserve distributional support while promoting goal satisfaction—even with few denoising steps.
>
> ## 5. Theoretical Contribution
>
> The theoretical motivation for Stein score-based selection is detailed in our response to Reviewer UvdB, point 4.
>
> We also see promising directions for further theoretical development. One avenue is to formalize the diversity gain induced by our population mechanism. We conjecture that in the limit of many denoising steps, population-based selection approximates the effect of increasing the noise level $ \gamma $, thus promoting broader exploration. A formal proof of this claim is left for future work.
>
> To the best of our knowledge, there is no theoretical description of the noise injection scale distortion in the current literature. Partial results are likely achievable in the coming week such as distribution support preservation using the line of argumentation of Pidstrigach's "Score-Based Generative Models Detect Manifolds".
>
> ## 6. Quality
> We will conduct a thorough proofreading and will improve the overall writing quality and clarity of our paper.

---

> > ### Comment · Reviewer_ptGF · 2025-08-06
> >
> > I appreciate the detailed response from the authors. My concerns have been addressed.

---

### Official Review · Reviewer_UvdB · 2025-06-30

**Clarity:** 3
**Significance:** 3
**Originality:** 3
**Rating:** 4
**Confidence:** 3

**Summary:**

The authors propose a new population-based sampling strategy for denoising process in embodied AI tasks. The method filter denoising samples based on an OOD score to reduce the injected noise when denoising. The authors show improvement on both score and clipping frequency using several diffusion models.

**Questions:**

1. One of the major problem is: The clip operation in Eq.2 is the foundation of many following analysis. But I’m not sure whether this clipping is necessary for diffusion policy / flow-matching policy. Especially, for middle $t$, is it necessary to keep the clipped item between the threshold? I’ll be interesting to see some analysis like Fig3 but without clipping. How will the raw diffusion policy perform? Will the proposed method still work?

2. For results in Figure 3, it seems like less noise scale will in general leads to better score. Those yellow points with 0 noise injection scale usually yield best results, which seems no tradeoff at all. Is it still necessary to use flexible stochastic modeling method like diffusion model to learn the policy? Or it will be enough to learn through some deterministic policy network?

3. Can the authors further explain the "tradeoff conditioning complexity for distribution complexity" in L256?

4. It would be better to have some theoretical analysis.

**Ethical Concerns:**

["NO or VERY MINOR ethics concerns only"]

**Final Justification:**

This paper provides an interesting view into diffusion process and seems this method is able to serve as a trick to apply to different tasks.

**Quality:**

2

**Strengths And Weaknesses:**

Strengths:

- In general, I like the observations and analysis as illustrated in Section 2. It has an interesting topic with a clean method which is verified on several different tasks.

- The method itself is simple and seems to be helpful, which help people to gain more insights on what's going on on the diffusion policy.

Weaknesses:

- Currently, the analysis and some claims are somehow seem to be subjective and intuitive. It would be helpful if the authors could provide some theoretical analysis for this method.

- See questions below.

---

> ### Author Rebuttal · Authors · 2025-07-31
>
> We thank the reviewer for their comments. We appreciate that they find the topic interesting and our method well established. Below we address the points raised in the review.
>
> ## 1. Necessity of Clipping in Diffusion
>
> Each DDPM denoising step predicts a fully **denoised** sample, which is then re-noised at a lower noise level. Clipping is applied only to this fully denoised sample prior to re-noising. Its purpose is twofold: to enforce bounded action constraints and to avoid numerical instabilities at high noise levels—particularly with schedules like cosine. From this perspective, clipping may be viewed as a form of *universal guidance* (Bansal et al. "Universal Guidance for Diffusion Models") with a "potential well" guidance.
>
> While such clipping is helpful in standard DDPM settings, alternative noise parameterizations, such as the variance-exploding linear schedule used in EDM, are less prone to numerical instabilities. Notably, EDM avoids rescaling the original sample, ensuring that predictions remain within the convex hull of the data distribution. Similarly, flow-matching models inherently satisfy this constraint and do not require clipping.
>
> Although we considered a *clipping schedule* (loosening the constraint over time), we ultimately abandoned it: clipping is already a heuristic justified primarily by empirical results, and further engineering on top of it lacked theoretical grounding. Instead, our approach offers a principled solution by improving alignment between the denoising trajectory and the data distribution, without compounding ad-hoc mechanisms.
>
> ---
>
> ## 2. Is a Stochastic Policy Still Necessary if γ = 0 Performs Best?
>
> While the denoising process may become deterministic as γ → 0, the policy remains stochastic due to the random initialization from Gaussian noise. This setting mirrors standard DDIM and Flow Matching sampling: randomness exists only at the start, and is then followed by a deterministic process.
>
> Moreover, stochasticity is essential *during training*, even if the final policy is effectively deterministic at inference time. Stochastic training encourages *mode coverage*, which helps prevent mode collapse—a critical concern in robotics. For example, in a 2D pick-and-place task with obstacles, multiple disjoint paths may exist. If the obstacle changes location, a deterministic policy may fail catastrophically if it relies on a single, now-infeasible mode. A stochastic policy can maintain robustness by supporting multiple valid behaviors, from which guidance or downstream selection mechanisms can choose.
>
> ---
>
> ## 3. Tradeoff Between Conditioning Complexity and Distribution Complexity (L256)
>
> We address this fully in our response to Reviewer fdQZ, Point 2, under the "Horizon Tradeoff Hypothesis." In brief: shorter horizons simplify the output distribution but complicate conditioning (as the model must infer context from a single observation). Longer horizons ease conditioning (since the observation can often be assumed to be the initial state) but require modeling a more complex multi-step distribution. The intermediate horizon regime thus suffers from the compounded difficulties of both.
>
> ---
>
> ## 4. On the Lack of Theoretical Analysis
>
> We acknowledge the value of theory and are actively working toward formal guarantees. A first result we aim to establish is a bound of the form:
>
> $$
> \mathbb{E}_{t, x_t} \left( \left. \| \epsilon^\theta(t, x_t) - \epsilon(t, x_t) \|  \right|  \| \epsilon(t, x_t) \| \right) \leq C(\|\epsilon\|) \cdot \mathcal{L},
> $$
>
> where $ \mathcal{L} $ is the training loss and $ C(\|\epsilon\|) $ is an explicit *increasing* function vanishing at zero. This would characterize how the denoising error increases with noise magnitude, supporting the intuition that OoD predictions at high noise lead to degraded performance.  Our goal is to include the full results in the paper.
>
> A more ambitious goal is to relate the expected sample density under the learned distribution to the Stein score norm, potentially bounding it from below by a decreasing function of the score norm. This remains speculative, but we are exploring this direction.
>
> ## 5. Quality
> We will conduct a thorough proofreading and will improve the overall writing quality and clarity of our paper.

---

> > ### Comment · Reviewer_UvdB · 2025-08-04
> >
> > Thank the authors for the response. Most of my questions are addressed.

---

### Official Review · Reviewer_fdQZ · 2025-07-03

**Clarity:** 3
**Significance:** 3
**Originality:** 3
**Rating:** 5
**Confidence:** 4

**Summary:**

This paper introduces the Genetic Diffusion Policy (GDP), a novel method to accelerate the inference of diffusion models for robotic manipulation tasks. The authors identify that standard diffusion policies, largely adapted from image generation, are inefficient for robotics due to a computationally expensive, multi-step denoising process. A key issue identified is "clipping," a heuristic that forces predicted actions into a valid range but inadvertently pushes intermediate steps out-of-distribution (OoD), degrading performance and wasting iterations. By tailoring the denoising process to the low-dimensional nature of robotic action spaces, the authors show that performance can be improved with far fewer steps.

To address these challenges, GDP employs a population-based genetic algorithm during the denoising process. Instead of a single denoising path, GDP evolves a population of potential action sequences, using a fitness score to select for in-distribution trajectories and discard those prone to clipping errors. This genetic selection mechanism enhances both sample quality and stability, particularly for very low step counts. The method is evaluated on 14 tasks from D4RL and Robomimic, demonstrating that it can match or improve performance by up to 20% while reducing the number of required neural function evaluations to as few as two, significantly outperforming baselines like standard DDPM and Shortcut models.

**Questions:**

Overall, this is a promising paper with strong empirical results. The core argument regarding clipping-induced denoising defects is well-supported by the experiments and provides a clear, intuitive explanation for performance degradation in standard diffusion policies. The proposed Genetic Diffusion Policy (GDP) demonstrates impressive performance, especially in the two-step inference regime. However, the paper could be strengthened by addressing the following points:

**1. Clarification on Baselines and Comparisons:**

* **DDIM as a Benchmark:** While the paper mentions that DDIM did not perform well in experiments, it is a standard baseline for fast sampling. For a more complete comparison, could the authors include DDIM's performance in the main results tables (e.g., Table 1), even if it is suboptimal? This would provide readers with a direct comparison against a widely-known method.
* **Shortcut Model Analysis:** The Shortcut model is presented as a key competitor.
    * Does the noise scaling factor (`γ`) apply to the Shortcut model's inference, or is it only tunable for DDPM and GDP?
    * Figure 8 shows that the Shortcut model's performance is robust across different iteration steps but significantly lower than GDP's. Could the authors provide more insight into why this might be the case?
    * Was the Shortcut model benchmarked on the Robomimic tasks? The results are shown for Adroit Hand tasks but seem to be missing for Robomimic.

**2. Deeper Explanation of Key Concepts:**

* **The Horizon Tradeoff:** The authors hypothesize that intermediate horizons are more difficult due to a tradeoff between "conditioning complexity" and "distribution complexity". This is an interesting point that would benefit from more detail. Could the authors elaborate on this concept? For example, what defines these two complexities in the context of the robotic tasks, and how does their interplay lead to the observed performance dip?
* **The Role of the Genetic Algorithm:** The "Genetic Denoising" process is central to the paper's contribution. However, the current implementation consists of selection and duplication, without crossover or mutation. To better isolate the benefit of the population-based approach, it would be helpful to understand its specific role. How does multinomial selection and duplication of a population compare to a simpler heuristic, such as calculating the fitness score for all samples, selecting the single best one, and proceeding with only that trajectory? An ablation study on this point would clarify whether the population diversity itself is critical, or if the main benefit comes from simply identifying and repeating a single "good" sample.

**Ethical Concerns:**

["NO or VERY MINOR ethics concerns only"]

**Limitations:**

### Scope and Generalizability

The most significant limitation is that the method's core principles are fundamentally tied to the specific characteristics of **low-dimensional robotic action spaces**. The very premise that reducing noise and inference steps is beneficial runs counter to findings in high-dimensional domains like image generation. This confines the paper's impact, making it a specialized solution for Embodied AI rather than a general advancement for diffusion models.

This limitation extends to scalability within robotics itself. The experiments are conducted on tasks with action spaces of around 30 dimensions. It remains an open question how GDP would perform on more complex robotic platforms, such as humanoids, where action spaces can involve hundreds of dimensions. As dimensionality increases, the "curse of dimensionality" could render the fitness heuristics less effective and the population-based search far less efficient, potentially reintroducing the very performance issues the method aims to solve.


### Practical and Computational Considerations

While the paper focuses on reducing the number of neural function evaluations (NFE) to lower latency, it doesn't fully address the **overall computational overhead** of GDP.
* **Memory and Parallelism:** Maintaining a population of `P` samples increases memory usage by a factor of `P`. More importantly, it requires `P` parallel evaluations of the noise model at each step before selection can occur. The total wall-clock time of a 2-step GDP with a population of 16 or 32 might be comparable to or even exceed that of a standard 5 or 10-step DDPM on certain hardware. This practical tradeoff is not quantified.
* **Hyperparameter Sensitivity:** GDP introduces several new, potentially sensitive hyperparameters, including population size, survival number, and the fitness temperature `T`. The paper mentions using a "coarse parameter grid," which suggests that finding an optimal configuration for a new task may require a non-trivial and expensive tuning process. A robust method should ideally be less sensitive to such additions.

---

### Methodological Dependencies

The success of GDP is critically dependent on two external factors that limit its use as a "plug-and-play" solution.
1.  **Reliance on a Strong Base Model:** The method is not a panacea for poor policies. The authors note it performs best when the underlying diffusion model is already well-trained and the task is sufficiently complex. This means GDP acts as an **optimizer for an already competent policy**, not a tool to salvage a poorly trained one. It cannot fix fundamental issues in the model's ability to understand the environment from its observations.
2.  **Dependence on the Fitness Heuristic:** The entire genetic selection process hinges on the assumption that the fitness score (e.g., `φ_clip`) is a reliable proxy for sample quality. While the clipping-based score is intuitive and works well for the tasks presented, it is still a heuristic. For different robot morphologies, task objectives, or action representations, this heuristic might fail. If the fitness score does not accurately reflect proximity to the true data manifold, the genetic selection could be ineffective or even detrimental, actively selecting for suboptimal trajectories. The method's success is therefore fundamentally coupled to the quality of this handcrafted heuristic.

**Paper Formatting Concerns:**

None from my side

**Quality:**

3

**Strengths And Weaknesses:**

### Strengths of the Paper

* **Insightful Problem Formulation and Diagnosis:** The paper's primary strength is its clear diagnosis of a core issue plaguing diffusion policies in robotics: **clipping-induced denoising defects**. Instead of just proposing a faster method, the authors identify a root cause—the distributional mismatch created by clipping early in the denoising process—and build their solution around it. The direct correlation shown between clipping frequency and poor performance provides strong, intuitive evidence for their central claim.
* **Comprehensive and Rigorous Empirical Evaluation:** The claims are supported by an extensive set of experiments. The evaluation spans 14 different tasks across two major benchmarks (D4RL and Robomimic), multiple action horizons, and a wide range of inference budgets. The scale of the evaluation, with over 2 million runs and up to 500 seeds for some tasks, lends significant statistical weight to the findings and demonstrates the method's robustness.
* **Practicality and Impact on Off-the-Shelf Models:** A major advantage of the proposed approach is its practicality. The Genetic Diffusion Policy (GDP) and the simpler tweaks (reducing noise, adjusting the schedule) can be applied to **existing, pre-trained diffusion policies without any architectural changes or retraining**. This significantly lowers the barrier to adoption and means that performance gains can be realized on already-developed models, as demonstrated by their 2-step GDP outperforming fully iterated baselines.
* **Novelty of the Genetic Denoising Approach:** The paper introduces a novel application of a population-based sampling strategy to accelerate diffusion models. While genetic algorithms are well-established, their use as a selection mechanism within the denoising process to filter trajectories based on an OoD score is a creative and previously unexplored idea in this context. This opens a new research direction for improving diffusion samplers.

### Weaknesses and Opportunities for Improvement

* **Depth of the "Genetic" Contribution:** The term "Genetic Diffusion Policy" may slightly oversell the mechanism, which currently consists of **selection and duplication without the canonical genetic operators of crossover and mutation**. The paper does not provide an ablation study to demonstrate that the population-based multinomial selection is superior to a simpler heuristic (e.g., a greedy approach that selects the single "best" trajectory at each step and proceeds). Such an analysis would be crucial to justify the additional computational cost of maintaining a population and to clarify the specific role that population diversity plays in the final performance.
* **Nuances in Baseline Comparisons:** The treatment of established baselines could be more thorough.
    * **DDIM:** This model is a cornerstone of fast diffusion sampling but is dismissed relatively quickly in the text as not performing well. For a more complete scientific record, including DDIM's scores in the main results tables would provide a valuable, direct comparison for the reader, even if the performance is poor.
    * **Shortcut Models:** The analysis of Shortcut models feels incomplete. Their performance is shown to be stable but suboptimal, yet the reason for this is not deeply explored. Furthermore, the comparison is only performed on the Adroit Hand tasks, leaving an open question about how GDP compares to Shortcut on the Robomimic benchmark.
* **Under-Explored Hypotheses:** The paper presents several interesting hypotheses that are not fully substantiated. The most notable is the "horizon issue," where the authors suggest performance dips at intermediate horizons due to a tradeoff between conditioning complexity and distribution complexity. While plausible, this is presented as speculation without supporting analysis. The paper would be stronger if this claim were either supported with further evidence or explicitly framed as a direction for future work.
* **Generalizability Across Task Types:** The paper notes a clear difference in performance and improvement margins between the Adroit Hand and Robomimic tasks, hypothesizing this is due to the latter being more bottlenecked by the model's conditioning rather than the denoising process. This is a key point regarding the limitations and applicability of GDP. Exploring this difference further would provide valuable insight into which types of robotic manipulation problems stand to benefit most from this and similar denoising-focused improvements.

---

> ### Author Rebuttal · Authors · 2025-07-31
>
> We thank the reviewer for the thoughtful and constructive feedback. We appreciate the precision provided and the points raised. Below, we address the reviewer’s concerns in detail.
>
> ---
>
> ## 1. Weaknesses and Clarifications
>
> ### Generalizability Across Task Types
>
> We agree with the reviewer’s observation. Our intuition is that for tasks like those from Robomimic, the conditional action distributions are simpler and more localized. As our method enhances the denoising process but does not modify the base model, its benefit is limited when the target distribution is easily covered, even with a few steps. We infer that the limiting factor for the Robomimic tasks is a conditioning bottleneck.  The impact of out-of-distribution (OoD) trajectories is then smaller, leading to limited performance gains of our method despite latency reduction. While we did not formally validate this hypothesis, it is consistent with the empirical trends observed across all Robomimic tasks.
>
> ---
>
> ## 2. Baseline Comparisons
>
> ### DDIM Scores
>
> We added the DDIM results, see point 4 of the rebuttal to reviewer PHeJ.
>
> ### Shortcut Model and γ
>
> Shortcut is a self-consistency variant of Flow-Matching (FM), as such it does not inject noise during inference. Stochasticity only comes from the choice of starting point in latent space. The $\gamma$ parameter has no direct equivalent but may be related to the standard deviation of the latent space normal distribution during inference. We did not experiment with this possibility.
> Guo et al.'s "Variational Rectified Flow Matching" introduces stochasticity along the inference process via VAE step-model; however, we did not find an adequate equivalent to $\gamma$. Finally, combining both self-consistency (shortcut FM) and distributional step-model (variational FM) would likely motivate a publication on its own.
>
> ### Shortcut Performance on Adroit
>
> Shortcut models require the network to learn both a vector field and its integral. With a fixed architecture (same UNet backbone), this leads to capacity bottlenecks that likely explain the inferior performance observed.
>
> ### Shortcut on Robomimic
>
> We agree that comparing the shortcut with the 2-step GDP is relevant for Robomimic tasks; however, no shortcut model checkpoints were publicly available for Robomimic tasks, so the comparison would be unfair to the off-the-shelf diffusion models we benchmarked.
>
> ---
>
> ## 3. Conceptual Clarifications
>
>
> ### Horizon Tradeoff Hypothesis
>
> Thank you for highlighting this point. A thorough analysis of the horizon-dependence was initially intended to be included in the present work. However, we concluded that a proper treatment reveals deeper phenomena that merit a separate follow-up article. Below, we summarize our current insights.
>
> Empirically, we observe three distinct regimes based on the action horizon $ \mathbf{h_\mathcal{A}} $:
>
> - $ \mathbf{h_\mathcal{A}} \leq 5 $: **Low performance**
> - $ 24 \leq \mathbf{h_\mathcal{A}} \leq 50$: **High performance, then decreasing**
> - $ \mathbf{h_\mathcal{A}} \geq 100 $: **High performance resumes**
>
> We hypothesize that these regimes arise from competing factors:
>
> 1. **For very short horizons** (< 5): the action distributions are heavily overlapping and "blob-like", leading to mode confusion. The model struggles to disambiguate modes due to the compressed embedding space. Increasing the horizon expands the dimensionality of the action sequence, which helps disambiguate modes and better leverage conditioning.
>
> 2. **Distribution Complexity**: As the horizon increases, the model must represent more complex and multi-modal action distributions. Fortunately, diffusion models are expressive enough to accommodate this growing complexity.
>
> 3. **Conditioning Complexity**: Shorter horizons also require the model to infer contextual information (e.g., temporal position) from the observation, which makes conditioning harder. For example, in short-horizon settings, the model must deduce which timestep is being referenced based solely on the observation. In contrast, long-horizon models can typically assume that the observation corresponds to the first timestep.
>
> Hence, performance is governed by a trade-off between **distribution complexity** (which increases with horizon) and **conditioning complexity** (which decreases with horizon). We believe that the performance drop observed in the intermediate horizon regime $ 50 \leq \mathbf{h_\mathcal{A}} \leq 100 $ results from both complexities being simultaneously high.
>
> To further analyze this phenomenon, we performed a preliminary investigation of the structure of the learned action distributions. Specifically, we estimated:
> - the **intrinsic manifold dimension**,
> - the **intrinsic linear dimension**, and
> - the **number of distributional modes**.
>
> The ratio between the intrinsic linear and manifold dimensions offers a geometric measure of complexity, which complements standard mode-counting approaches.
>
> Looking forward, we aim to explore more principled **Wasserstein-based complexity measures**, which can capture both the shape and spread of distributions:
> - **Expected Wasserstein Distance to Gaussian** (as a proxy for distributional complexity),
> - **Wasserstein Barycenter Distance distance to Gaussian** (another proxy for distribution already complexity), and
> - **Wasserstein Variance** (quantifying the distributionnal diversity across conditioning).
>
>
>
>
> ### Role of the Genetic Algorithm
>
> Our goal was to validate the utility of population-based denoising without introducing algorithmic complexity. We did run comparative experiments
>  Top-k vs. Multinomial Selection: At 2 steps, top-k outperforms multinomial slightly. However, it quickly collapses diversity in multi-step settings, leading to mode collapse.
>
> While a greedy selection heuristic could achieve similar performance in the very low-step regime, it would fail to scale beyond that. The population-based design is essential for maintaining robustness as the denoising depth increases.
> If diversity is what the current task requires, more sophisticated population-based metaheuristics like
> Particle Swarm Optimization or Ant Colony Optimization might be better suited and could be applied, but we leave such an exploration to future work.
>
> ---
>
> ## 4. Practical and Computational Considerations
>
> ### Memory and Wall-Clock Cost
>
> We measured the step-wise overhead for GDP using a standard RTX 3080 GPU without specific optimization beyond calling the model batch-wise on the population.
>
> |Pop. Size |NFE cost ($\mu$s)| Step cost  ($\mu$s)|  Overhead ratio|
> |:----:|:----:|:----:|:----:|
> |1 (DDPM)|3800|200|1|
> |8|3800|500|1.075|
> |16|4000|800|1.2|
> |32|4500|1500|1.5|b
> |64|5500|2400|1.975|
>
> The relatively low cost is due to the under-utilization of the GPU computation and memory. As the model grows bigger, the population may be more constrained. We argue the following.
> - In production, the observation embedding may represent a significant portion of the computational budget but since it can be shared across the population, its impact on the overhead is limited.
> - Training is typically done with a batch size larger than one; thus, assuming training and inference are done on the same hardware, inference is likely to leave room for a non-trivial population.
> - Fully-fused models, weight quantization, and other low-level optimization would allow for a larger batch size (hence a population) during inference.
>
> These points will be discussed in a dedicated section in the appendix.
>
>
> ### Hyperparameter Sensitivity
>
> Our coarse grid search served both as an ablation and robustness check. We found GDP to be relatively insensitive to population sizes in [8, 32] and temperature T in [1, 1000]. Increasing temperatures caused performance to vary from optimal at low temperatures to vanilla DDPM at high temperatures. While additional hyperparameters are introduced, we show that good performance is obtainable across a wide range without fine-tuning. That said, we acknowledge that future work should explore more principled tuning strategies or adaptive methods.
>
> ---
>
> ## 5. Limitations
>
> ### Dependence on Fitness Score
>
> Indeed, the success of GDP depends on the choice of fitness function. For robotic tasks with saturation constraints, our clipping-based score is effective. For tasks where clipping is rare (e.g., end-effector control, image domains), this heuristic is less useful. Still, whatever the task, Stein-based scores are always available and valid candidates. Furthermore, we note that it may be possible to learn a task-specific fitness function in a supervised or self-supervised manner, which is a promising direction for future work.
>
> ### Scalability to High-Dimensional Action Spaces
>
> While our action spaces have ~30 DoFs, our experiments span horizons up to 200, leading to effective dimensions over 6000. This is higher than image datasets like CIFAR. Furthermore, many humanoid benchmarks such as HumanoidBench have ≤61 DoFs, keeping GDP within practical scaling bounds for whole-body control tasks.
>
> We will note this point in the limitations.
>
>
> ## 6. Quality
>  We will conduct a thorough proofreading and will improve the overall writing quality and clarity of our paper.

---

> > ### Comment · Reviewer_fdQZ · 2025-08-05
> > **Rebuttal Received**
> >
> > I would like to thank the authors for their comprehensive and thoughtful rebuttal. By providing new experimental results, including a crucial ablation study, and clarifying the justification for their design choices, they have addressed the points raised in my initial review.

---

### Official Review · Reviewer_PHeJ · 2025-07-03

**Clarity:** 2
**Significance:** 2
**Originality:** 3
**Rating:** 4
**Confidence:** 4

**Summary:**

This paper investigates the dynamics of diffusion policy to optimize the sampling process. The authors find that by decreasing the noise injected + having a shorter denoising schedule, the sampling pipeline could be greatly accelerated. They also claim that poor performance is related to the clipping during the sampling process, and propose a genetic algorithm for sampling. They evaluated their hypothesis on various robotics tasks.

**Questions:**

1. For the experiments section, the paper lacks details in the experiment setup. For example, shortcut models require training, but no details were given in how the process was done.
2. In the experiments section, what is

    > the the number of diffusion steps in {1,2,⋯,10} ∪ {20,30,⋯,100}
    >

    Are many models being retrained with different numbers of diffusion steps? Or does this refer to the number of denoising steps, and the base model is always trained with 100 diffusion steps? Additionally, what denoising schedule is selected?

3. Why not include DDIM results in the experiments table? DDIM was included as part of the experiment, but the results (explicit scores) are never reported.
4. For the Robomimic experiments, why are DDPM results missing in the last section, when the number of steps is 2?
5. For the DDPM experiment entries, why is the noise injection only taken at extreme values (1 and 0)? what about the intermediate ones?
6. For the proposed GDP method, there are two OoD score measurements proposed, which type did the experiments use?
7. It’s unclear to me what figures 5 and 6 are depicting. The experiments have 4 dimensions (steps, noise scale, horizon, and method). Both of these are 2D graphs; what’s happening to the other two dimensions? Are they specific runs? Averaged over the other two dimensions?

**Ethical Concerns:**

["NO or VERY MINOR ethics concerns only"]

**Final Justification:**

My initial low score was due to the low quality presentation of the experiments, methods, and results. During the rebuttal period, most of the answers were addressed by the authors, and they promised to improve and address my concerns in the final version. Thus, I have increased my score on the paper.

**Limitations:**

Yes.

**Paper Formatting Concerns:**

None.

**Quality:**

2

**Strengths And Weaknesses:**

Strength:

1. Proposes a new population-based algorithm for effective sampling under limited diffusion timesteps.
2. Interesting finding that clipping leads to poor performance.

Weakness:

1. Experiments and methods are not clear. (see questions)
2. Limited algorithm evaluation. Missing ablation studies to show the effectiveness of their method.

---

> ### Author Rebuttal · Authors · 2025-07-30
>
> We thank the reviewer for their constructive comments and insightful questions. The  requested experiments were omitted due to space limitation and the focus of the paper. However, we agree with the reviewer that they are valuable and therefore added them as requested (see Point 5).
>
>
>
> ## 1. Experimental details - Shortcut model training
> We re-implemented the shortcut model from Algorithm 1 in Frans et al., 2024 in PyTorch, using their official GitHub repository (which is in Jax). For each (task, horizon) pair, we trained 10 models via random hyperparameter search. Each reported score on a given task is the best-performing out of all 10 models. All models use the same 65M-parameter UNet backbone to ensure fair comparison.
>
> ## 2. Models Retraining
>
> All models were trained once using 100 denoising steps (128 for the shortcut model). At inference, we vary the number of denoising steps using the cosine schedule from Diffusion Policy. We also experimented with linear schedule and found that the impact of our method was similar but the performance of linear schedule models was significantly lower compared to cosine schedule both for baseline DDPM and GDP. We therefore decided to focus on cosine schedule.
>
> ## 3. Clarification on Step Range in Experiments
>
> The set {1, 2, ..., 10} ∪ {20, 30, ..., 100} refers to inference-time denoising steps only. All base models were trained using the full 100-step schedule (or 128 for shortcuts), and inference was conducted by subsampling the training schedule (Please refer to Equation (2) and lines 85–86 in the paper).
>
> ## 4. DDIM results
>
> The values for DDIM are compiled in the table below and integrated in tables 1 and 2 of the submission.
> |Name|100 steps|5 steps|2 steps|
> |----|----|----|----|
> |lift.ph|0.998|0.998|0.998|
> |lift.mh|0.998|0.998|1|
> |can.ph|0.982|0.954|0.982|
> |can.mh|0.97|0.956|0.966|
> |square.ph|0.928|0.898|0.922|
> |square.mh|0.846|0.826|0.842|
> |transport.ph|0.846|0.808|0.821|
> |transport.mh|0.606|0.606|0.604|
> |toolhang|0.514|0.51|0.48|
> |-|-|-|-|
> |Hammer|0.70|0.71|0.76|
> |Relocate|0.38|0.38|0.42|
> |Pen|0.50|0.70|0.73|
> |Door|0.83|0.81|0.95|
>
> ## 5. Thorough ablation
>
> Our method modifies the denoising process in three ways: adapting inference schedule, reducing noise injection scale and genetic denoising.  For completeness, we added the missing lines to tables 1 and 2 of the submission.  For 2-step we provide the normalized scores for
> - DDPM,
> - DDPM + adapted schedule,
> - DDPM + adapted schedule + reduced noise,
> - DDPM + adapted schedule + reduced noise + GDP.
>
> 2-step DDPM alone always yields minimal score as shown in the table below,
>
> | 2-step, adapted schedule    | gamma=0 | gamma=1 |
> |----|----|----|
> |Hammer|0.95|0.87|
> |Pen|0.75|0.74|
> |Relocate|0.74|0.64|
> |Door|1.0|0.97|
> | ------------ | ------------------ | ------------------|
> | lift.ph      | 0.998              | 0.998             |
> | lift.mh      | 0.995              | 0.998             |
> | can.ph       | 0.989              | 0.984             |
> | can.mh       | 0.968               | 0.970            |
> | square.ph    | 0.919              | 0.934             |
> | square.mh    | 0.826              | 0.845             |
> | transport.ph | 0.786              | 0.818             |
> | transport.mh | 0.595              | 0.578             |
> | toolhang     | 0.481              | 0.502             |
>
> Note that on Robomimic, the difference between $\gamma=0$ and $\gamma=1$ scores is not statistically signifiant as the difference is within 1-sigma.
>
> ## 6. Noise injection scales
>
> Figure 3 and Appendix's Figure 9 include intermediate $\gamma$ values. We chose not to include them in the tables for brevity, as $\gamma= 0$ consistently yielded the best results.
>
> ## 7.  Which OoD score was used for GdP experiments
>
> Both Stein and clip-based scores were tested. The Stein score consistently performed better, both in reducing clipping and improving success rate. Thus, we report results using the Stein score.
>
> ## 8.  Figure 5 and 6
>
> Figures 5 and 6 are averaged over the remaining two dimensions (e.g., horizons and tasks). We updated the captions to clarify this.
>
> ## 9. Quality
> We acknowledge that the current writing should be improved and we are working to improve writing quality and overall clarity.

---

> > ### Comment · Reviewer_PHeJ · 2025-08-05
> >
> > Thank you for the added clarifications. I am willing to increase my score, considering the writing and presentation will be improved in the final submission.

---

### Decision · Program_Chairs · 2025-09-17

**Decision:**

Accept (poster)

**Comment:**

This paper proposes a population-based denoising strategy -- in this case a genetic algorithm -- to generate actions for robotic manipulation tasks. The authors identify a key failure mode of action generation -- clipping -- and propose their population-based method as a successful fix, achieving improved performance in D4RL and Robomimic. The reviewers found several shortcomings in the original draft -- notably on the lack of support for some of the story and missing ablations -- but these have been well-addressed in the rebuttal discussion.

I therefore recommend the paper is accepted as a poster.